# Nitrogen- and Halogen-Free Multifunctional Polymer-Directed Fabrication of Aluminum-Rich Hierarchical MFI Zeolites

**DOI:** 10.3390/nano12101633

**Published:** 2022-05-11

**Authors:** Yuan Hu, Chan Wang, Tiesen Li, Xiaojun Bao, Yuanyuan Yue

**Affiliations:** 1National Engineering Research Center of Chemical Fertilizer Catalyst, College of Chemical Engineering, Fuzhou University, Fuzhou 350116, China; huyuankuaile@126.com (Y.H.); chan.wang@fzu.edu.cn (C.W.); litiesen@fzu.edu.cn (T.L.); baoxj@fzu.edu.cn (X.B.); 2Qingyuan Innovation Laboratory, Quanzhou 362801, China

**Keywords:** hierarchical ZSM-5 zeolites, aluminum-rich, nitrogen- and halogen-free polymer, waste plastics cracking

## Abstract

Aluminum-rich hierarchical MFI-type zeolites with high acidic-site density exhibit excellent activity and selectivity in bulky molecule-involved reactions. However, it is challenging to develop a facile and environmentally benign method for fabricating them. Herein, we employ a polymer that does not contain nitrogen and halogen elements to successfully synthesize aluminum-rich hierarchical ZSM-5 zeolite with a Si/Al ratio of 8 and a significant number of mesopores comprised of oriented-assembled nanocrystals. It is demonstrated that the nitrogen- and halogen-free polymer is instrumental in the formation of the ZSM-5 zeolite by serving as a template for constructing the hierarchical micro/mesoporous structure. Moreover, this polymer also acts as a crystal growth modifier to form a single-crystalline zeolite. Notably, the resultant zeolite shows a better catalytic performance in converting waste plastic into hydrocarbons than a commercial one. Our work enables the synthesis of high-quality hierarchical zeolites without requiring quaternary ammonium templates.

## 1. Introduction

ZSM-5 zeolite with an MFI framework has been widely applied as a promising catalyst in a variety of fields such as petroleum refining and petrochemical industries because of its tunable acidity, hydrothermal stability, and uniform micropores of molecular dimensions [1,2,3]. Its catalytic performance mainly depends on the following two key factors: acidity and accessibility of the active sites. The acid sites of ZSM-5 are derived from the presence of protons balancing the negatively charged [AlO_4_]^−^ tetrahedra [4], which are often directly related to the Al content in the zeolite framework, i.e., the lower Si/Al ratio, the more acid sites. Therefore, aluminum-rich ZSM-5 zeolites should provide a high acid-site density increasing the catalytic activity. However, most of the acid sites in ZSM-5 zeolite are imbedded in the interior of microporous channels with sizes of 0.52–0.56 nm resulting in poor accessibility during catalytic reactions. Therefore, fabrication of ZSM-5 zeolites with high acidic-site density and improved accessibility to acid sites is of great importance.

ZSM-5 zeolite is a highly siliceous zeolite that can typically be synthesized in good quality with Si/Al ratios scoping from about 12 to an indefinitely high number. The fabrication of Al-rich ZSM-5 zeolites with Si/Al ratios lower than 12 and an enhanced concentration of Brønsted acid sites is of considerable interest but remains a great challenge [5]. Desilication in basic media results in only a marginal increase in the aluminum content. Thus, the framework aluminum content is usually increased during the synthesis process. Beck et al. [6] synthesized ZSM-5 with a Si/Al ratio of 7.5, but the sample was impure. When the aluminum content of the synthesis gel was increased, analcime and mordenite were formed as impurities [7]. Romannikov et al. [8] had synthesized ZSM-5 zeolite with a Si/Al ratio of 11; however, a part of an octahedrally coordinated Al species was observed in the zeolite sample. More recently, Pereira et al. [9] had successfully prepared ZSM-5 zeolites with a Si/Al ratio of 8 by using tetrapropylammonium hydroxide as the structure-directing agent and sugar cane residues as crystal-growth modifiers. Previous studies [9,10] have shown that templates play a crucial role in aluminum insertion during the synthesis of aluminum-containing zeolites. These templates containing hydroxy groups can chelate Al^3+^ [11,12], which favors a higher Al content in the final zeolite. Furthermore, apart from acidity, accessibility to acid sites is also crucial for the conventional microporous Al-rich ZSM-5 zeolite that suffers from diffusion limitation leading to rapid deactivation in the practical applications because of coke deposition. However, the accessibility to acid sites can be improved by creating mesopores in the microporous zeolites [13,14,15,16]. Post-synthetic desilication of Al-rich ZSM-5 zeolites (Si/Al ratio < 25) to generate additional mesopores is relatively unexploited because the high Al concentration in the framework hinders the extraction of silicon from the zeolite structure [17]. Bottom-up self-assembly methods using multifunctional hard [18] /soft [19,20,21,22,23,24,25,26] macro/mesoporogens to form aluminum-rich hierarchical ZSM-5 zeolites have been developed. However, these strategies normally employ nitrogen- and/or halogen-containing compounds as structure-directing agents or macro/mesoporogens [27,28,29] generating the emissions of toxic NO_x_ and/or HX into the environment during their removal process via high-temperature calcination. Therefore, the direct synthesis of aluminum-rich hierarchical zeolites via a nitrogen- and halogen-free route is highly desirable.

In this work, we demonstrate a novel multifunctional polymer template (OH-[CH_2_-C_6_H_10_-CH_2_-O-C(CH_3_)_2_-O]_78%_-[C_5_H_10_-O-C(CH_3_)_2_-O]_22%_-*_,_ denoted as PK3) to synthesize aluminum-rich hierarchical zeolites. The polymer can act as a dual-functional template to simultaneously generate zeolitic micropores and mesopores. It can also be an efficient zeolite growth modifier to effectively regulate the anisotropic growth rates of crystals, inducing the formation of single-crystalline zeolites. PK3 can bridge multiple crystal surfaces and fill in crystal-linkage spaces, thereby leading the formation of mesopores. More importantly, as a polymer decorated with hydroxyl groups, PK3 can also chelate Al^3+^, which favors a high Al content in the final zeolites. The obtained aluminum-rich single-crystalline hierarchical ZSM-5 zeolite exhibits higher catalytic activity than the conventional ZSM-5 zeolite in bulky molecule-involved reactions.

## 2. Materials and Methods

### 2.1. Materials

1,4-Cyclohexanedimethanol, 1,5-pentanediol, 2,2-dimethoxypropane, and p-toluene sulfonic acid (PTSA) were purchased from the Aladdin Reagent Company (Shanghai, China). Cyclohexane, ethyl acetate, triethylamine, and n-hexane were bought from Sinopharm Chemical Reagent Company (Shanghai, China) and distilled with calcium hydride. NaAlO_2_ (57.1% Al_2_O_3_ and 37.2% Na_2_O) and NaOH (96 wt%) were gained from Sinopharm Chemical Reagent Company (Shanghai, China) and fumed silica was obtained from the Shanghai Tengmin Industrial Company (Shanghai, China). Commercial H type ZSM-5 zeolite with a Si/Al molar ratio of 12, denoted as C-Z5, was acquired from Nankai University Catalyst Co., Ltd. (Tianjin, China). Low-density polyethylene (LDPE) powder used as a model feedstock in the cracking reaction was provided by Alfa Aesar (Stock #42607) (Tianjin, China).

### 2.2. Synthesis of Ketal Polymer

PK3 was synthesized according to the reference procedure [30]. First, 35 mmol of 1,4-cyclohexanedimethanol and 9 mmol of 1,5-pentanediol were dissolved in 30 mL of cyclohexane and heated at 100 °C for 0.5 h under magnetic stirring. Subsequently, 0.02 mmol of PTSA dissolved in 4 mL of ethyl acetate was added. After the ethyl acetate was evaporated, 44 mmol of DMP was added to start the reaction. After reaction for 6 h, to make up for the DMP and cyclohexane that had been boiled off an additional 3 mL of DMP and 5 mL of cyclohexane were subsequently added into the reaction mixture (this was repeated every 6 h for a total of five times). At last, 2 mL of triethylamine was added to stop the reaction. After quenching and purification, a solid product with the formula of OH-[CH_2_-C_6_H_10_-CH_2_-O-C(CH_3_)_2_-O]_78%_-[C_5_H_10_-O-C(CH_3_)_2_-O]_22%_-* and molecular weight of 3.9 KDa was obtained and designated as PK3. The molecular weight of the synthesized product could be tuned by the number of additions of DMP. The product obtained with a single addition of DMP was denoted as PK3-1 with a molecular weight of 0.36 KDa, PK3 ^1^H NMR (400 MHz, CDCl_3_, δ): 3.95–3.21 (m, 4H, CH_2_), 1.84 (s, 1.7H, CH), 2.21–1.45 (m, 8.2H, CH_2_), and 1.25 (s, 6H, CH_3_).

### 2.3. Synthesis of Z5-PK3

0.24 g of NaAlO_2_ and 0.32 g of NaOH were added to 27 mL of deionized water and stirred until the solution was clarified. Then 1.7 g of PK3 was added and remained under magnetic stirring for 0.5 h. Afterward, 1.9 g of fumed silica was added slowly to obtain a gel with a composition of 20 SiO_2_/1 Al_2_O_3_/3.42 Na_2_O/961.6 H_2_O/0.28 PK3. The resultant gel was agitated at 25 °C for 2 h and then transferred into a Teflon-lined stainless-steel autoclave that was placed in a homogeneous reactor at a rotational speed of 60 rpm and crystallized at 160 °C for 96 h. After crystallization, the obtained powder was filtered, washed with deionized water, dried at 100 °C for 12 h, and calcinated at 550 °C for 6 h to yield a Na-type ZSM-5 denoted as Na-Z5-PK3. Na-Z5-PK3 was further ion-exchanged with 1 M NH_4_NO_3_ solution at 80 °C for 7 h (this process was repeated two times), and then calcinated at 450 °C for 7 h to yield a H-type ZSM-5 denoted as H-Z5-PK3. PK3-1 was also applied in the synthesis of ZSM-5 zeolite by replacing PK3 with the similar synthesis composition used for synthesizing Z5-PK3, and the obtained product was marked as H-Z5-PK3-1. For comparison, another synthesis without the addition of PK3 was also carried out under the same conditions as those used for synthesizing Z5-PK3, and the obtained sample was labeled as Z5-PK3-free.

### 2.4. Characterization Techniques

^29^Si, ^13^C, and ^27^Al magic-angle spinning (MAS) and ^1^H NMR were performed using a Bruker Avance III/WB-600 spectrometer (Bruker Inc., Zurich, Switzerland). Solid-state two-dimensional (2D) ^23^Na {^1^H} heteronuclear correlation (HETCOR) NMR spectra were recorded with a ramped contact pulse of 3 ms. The morphologies of zeolites were detected by double-beam field emission scanning electron microscopy (SEM) using a Helios G4 CX/Helios G4 CX electron microscope (Thermo Scientific Inc., Waltham, MA, USA) equipped with an EDAX Inc. Octane Elect Super X-ray energy dispersive spectrometer (EDS) system. Transmission electron microscopy (TEM) images were obtained using a Tecnai F20 and Talos F200X (FEI Inc., Hillsboro, OR, USA) electron microscope operated at 200 kV. The relative crystallinities and phase purities of the as-prepared samples were determined by powder X-ray diffraction (XRD) using a Rigaku D-Max 2550 diffractometer (Bruker Inc., Karlsruhe, Germany) with Cu Kα radiation (λ = 1.5418 Å). Ar adsorption–desorption measurements were performed on a Micromeritics 2460 analyzer (Micromeritics Inc., Atlanta, GA, USA) at –186 °C after the samples were degassed at 300 °C under vacuum. Thermogravimetric analysis (TGA) was performed using a PerkinElmer STA 6000 instrument (PerkinElmer Inc., Waltham, MA, USA). Inductively coupled plasma (ICP) was carried out on a Perkin-Elmer Optima 3300 DV ICP instrument (PerkinElmer Inc., Waltham, MA, USA). The pyridinated zeolites were characterized by Fourier transform infrared spectroscopy (FTIR) performed on a Bruker Vertex 80 spectrometer (Bruker Inc., Karlsruhe, Germany) with a Harrick praying mantis diffuse reflection (DRIFTS) attachment and analyzed using OPUS 7.0 software.

### 2.5. Molecular Modeling

To study the adsorption of PK3 on the (010), (100), and (101) crystal planes of the synthesized MFI zeolites, umbrella sampling molecular dynamics (USMD) simulations were conducted using GROMACS-2020.6. The details of computation were according to the reference [31].

### 2.6. Catalytic Tests

The LDPE cracking reaction was conducted on TGA. The catalysts were mixed with LDPE powder at a catalyst/LDPE weight ratio of 1/3, and the mixture was then placed into the corundum crucible of the TGA instrument, and N_2_ at a flow rate of 50 mL min^−1^ was used to remove the air in the oven. The reactions were carried out in the TGA instrument by elevating the temperature to 600 °C at a ramp rate of 10 °C min^−1^ in N_2_. The conversion of LDPE was calculated from the weight loss during the reaction. To regenerate the zeolite catalyst and quantify the amount of coke after LDPE cracking, the spent sample was heated from 25 °C to 700 °C at a ramp rate of 10 °C min^−1^ in O_2_ (90 mL min^−1^). The regenerated sample was collected and weighed, then mixed with LDPE in a catalyst/LDPE weight ratio of 1/3 to continue the pyrolysis-regeneration cycles.

## 3. Results

### 3.1. Characterization of MFI Zeolites

The XRD patterns of H-Z5-PK3 and H-Z5-PK3-1 show peak at 2θ of 7.96°, 8.82°, 23.27°, 23.97°, and 24.43°, corresponding to (011), (200), (051), (033), and (133) reflections of MFI-topology, respectively, indicating formation of microporous zeolite ZSM-5 crystals. No other unidentified peaks except the MFI-type reflection peaks were observed (Figure 1), suggesting that the synthesized samples are highly pure and crystalline MFI-type zeolites. Figure 2 depicts the SEM and TEM images of H-Z5-PK3, which displays a xanthium sibiricum-like morphology composed of aggregated, ultrasmall nanoparticles with a size of ca. 90 nm (Figure 2a), and no amorphous components are found. The high-resolution TEM image in Figure 2c clearly shows that the lattice fringes extend over the entire crystal along consistent orientations, confirming the single-crystalline nature. Overlaying the MFI framework structure along the [010] direction in the higher-magnification images of the framework at the crystal periphery (Figure 2d) reveals an excellent match. The fast Fourier transform (FFT) diffraction pattern (Figure 2f) of the whole particle has clear and regular diffraction spots, indicating the [010] orientation of the nanocrystal. These nanocrystals are assembled in the same orientation, displaying oriented growth, and the entire particle has a single-crystal structure. The high-angle annular dark-field scanning transmission electron microscopy (HAADF-STEM) image (Figure 3) has good structural contrast and shows a somewhat three-dimensional distribution of the hierarchical structure, showing the coexistence of both micropores and intracrystal mesopores formed by the oriented assembly of nano-sized crystals. The coexistence of micropores and mesopores of H-Z5-PK3 can be further confirmed by the Ar adsorption–desorption isotherms. The steep rise uptake at a low relative pressure (P/P_0_ < 0.1) reveals the presence of micropores in H-Z5-PK3 (Figure 4a). At higher relative pressure, the combined features of type I and IV isotherms with a H3-type hysteresis loop (P/P_0_ that ranges from 0.80 to 0.99, Figure 4a) are observed. These are ascribed to the adsorption of Ar molecules onto the mesopores, which results from the intercrystalline voids. The Barrett–Joyner–Halenda (BJH) pore size distribution curve (Figure 4b) exhibits a large pore-size distribution in the range of 10–100 nm. Based on the data in Table 1, when comparing with a reference H-form ZSM-5 zeolite (C-Z5), H-Z5-PK3 shows an apparently larger BET surface area (421 vs. 345 m^2^/g) and a higher mesopore volume (0.16 vs. 0.03 cm^3^/g) owing to the high crystallinity and number of mesopores.

The chemical states of aluminum and silicon species in the Na- and H-type Z5-PK3 were further studied by ^27^Al and ^29^Si MAS NMR techniques, respectively. The ^27^Al MAS NMR spectra of Na-Z5-Pk3 shows a single sharp signal at ca. 55 ppm, while the resonance at 0 ppm is absent, indicating it consists of a tetrahedral framework Al-species (Al^IV^) (top in Figure 5a). After ion exchange and calcination, the spectrum of H-Z5-PK3 (bottom in Figure 5a) shows a small quantity of aluminum at 0 ppm that relates to the extra-framework Al species (Al^VI^). The ^29^Si MAS NMR spectra primarily include five resolved signals [32,33]: the peak at −98 ppm corresponds to a defective Q^2^(2Al) unit; the peak at −102 ppm corresponds to Q^3^(0Al) at the external surface of the zeolites; and the peaks between −106 and −109 ppm are attributed to Q^4^(1Al) Si species, that is, sites with one neighboring aluminum atom in the second coordination sphere. Those at −112 and −115 ppm belong to Q^4^(0Al) framework Si species. The relative distribution of these Si species as obtained by the deconvolution of the ^29^Si NMR spectra is summarized in Table 2. It shows that there are no existing defective Q^2^ units ([(HO)_2_Si(OT)_2_]) in all samples; the Na-Z5-PK3 consists of 92% Q^4^ units ([Si(OT)_4_]) and 8% Q^3^ units ([(HO)Si(OT)_3_], silanol species) (Table 2), and the H-Z5-PK3 consists of 91% Q^4^ units and 9% Q^3^ units. From the deconvolution of the ^29^Si MAS NMR spectrum (Figure 5b and Table 2), it can be obtained that the framework Si/Al ratio of the Na-Z5-PK3 is 7 and the H-Z5-PK3 is 8. From TEM-EDX mapping (Figure 3c,d), it shows the uniform distribution of both Al and Si, and there is no Al-enrichment zone in H-Z5-PK3. These results show that this nitrogen- and halogen-free polymer can successfully induce the synthesis of aluminum-rich single-crystalline hierarchical ZSM-5 zeolites with high purity.

### 3.2. Mechanism of Zeolite Growth

Na^+^ can be used as an inorganic template to direct the formation of MFI structure; however, in this work, the only use of Na^+^ results in a mordenite (MOR) phase with larger particle sizes (~180 nm) (Figure 1 and Figure 6). The solid-state ^13^C MAS NMR spectra (Figure 7a) of Na-Z5-PK3 exhibits the peaks associated with the PK3 species. This means that PK3 remains intact during the crystallization process and is incorporated into Na-Z5-PK3 because its flexible polymer chain allows it to fill the micropores. Site-specific interactions of the framework ^23^Na species with PK3 molecules were established by solid-state 2D HETCOR NMR spectra of Na-Z5-PK3 (Figure 7b). With the use of 2D NMR correlation spectra, mutual proximities or covalent connectivities of corresponding ^1^H, ^23^Na species can be manifested by correlated signal intensities [34]. 2D ^23^Na {^1^H} HETCOR NMR spectra of Z5-PK3 show two strong correlated 2D signal intensities, the ^23^Na signal at −5 ppm correlates with ^1^H signals at 2.2 and 4 ppm (Figure 7b) assigned to PK3 (3.95–3.21 (m, 4H, CH_2_) and 2.21–1.45 (m, 8.2H, CH_2_)), and are especially correlated to the ^1^H signal at 4 ppm originating from the PK3 coordinated to Na^+^ cations [35]. It can be concluded that Na^+^ collaborates with PK3 to direct the formation of ZSM-5 (the obtained product in the absence of Na^+^ is amorphous).

Macromolecules decorated with hydroxyl groups tend to effectively control the growth of crystals by tailoring the morphology and size of zeolites [36,37]. It is likely that PK3 with alcohol groups can form H-bonds with surface silanols, adsorb on crystal and/or precursor surfaces, and inhibit the growth by particle attachment through steric stabilization [38]. To verify the hypothesis that preferential interactions between PK3 and zeolite surfaces lead to the [010] orientation of zeolite nanocrystals, we probed the interaction of PK3 with different MFI surfaces by means of USMD simulations (Figure 8) to compute the adsorption free-energy profile F(z). As shown in Figure 8, the function F(z) is nearly constant for z beyond 0.5 nm for each surface, manifesting that the interactions between PK3 and ZSM-5 are short ranged. Obviously, the adsorption free energies of PK3 for the (010) and (101) surfaces of MFI are comparable and more favorable than those for the (100) surface (Figure 8). Previous studies demonstrated that the (010) surface was the most energetically favorable orientation among the three MFI orientations, (001), (010), and (101); thus, this surface is favorable for adsorption of organic additives [39]. Thus, PK3 preferentially binds to the MFI (010) surface through the hydrogen binding between alcohols on PK3 and the exposed ≡SiOH or ≡AlOH groups presented on the (010) surface. Crystal growth in the (010) direction is inhibited and in an identical orientation is obtained, while the surfaces of the crystals are bridged and filled with absorbed PK3, inducing the formation of (010)-oriented-assembly single crystals.

Z5-PK3-1 synthesized by using PK3-1 with molecular weight of 0.36 KDa as a template is comprised of spheroid-shaped ZSM-5 nanocrystals and does not exhibit noticeable anisotropy (Figure 1 and Figure 9). This phenomenon demonstrates that macromolecules tend to be more effective modifiers than small organic molecules because their proximal binding moieties can simultaneously bind to the crystal surfaces [38]. PK3 also functions as a “porogen” on the mesoscale, and the Ar adsorption–desorption isotherms and pore-size distribution curves (Figure 4 and Table 1) indicate that the mesopore volume and size of the MFI zeolite increase with the molecular weight of PK3. These mesopores are formed after the removal of the PK3 that occupies the interstitial spaces between the nanocrystals.

### 3.3. Catalytic Performance for LDPE Cracking

The catalytic performance of the synthesized hierarchical zeolites (H-Z5-PK3-1 and H-Z5-PK3) and the reference zeolite (C-Z5) for LDPE pyrolysis was evaluated. These zeolites have different pore structures and acid strengths (Table 1). It is obvious that H-Z5-PK3 with the highest acid amount efficiently converts LDPE to hydrocarbons at the lowest temperature (Table 3 and Figure 10a–c). Four consecutive pyrolysis-regeneration cycles were also carried out. During these tests, the spent catalysts were regenerated by calcination at 700 °C, then reused in subsequent reaction cycles [40,41]. Two different temperatures, T20 and T50, at which 20 wt% and 50 wt% of the LDPE is pyrolyzed to hydrocarbons, are compared to evaluate the performances of the various ZSM-5 zeolites. At T20, cracking occurred mainly at acid sites on the external surface. At T50, the degraded polymeric species diffused into the pores of the zeolite and further degraded, indicating that the interior acid sites are accessible [42]. H-Z5-PK3 and H-Z5-PK3-1 are nanocrystalline zeolites, which present a larger external surface area (Table 1), shorter diffusion path, and more accessible active sites than micron-sized C-Z5. The temperatures required to achieve conversion of 20 wt% and 50 wt% over H-Z5-PK3 and H-Z5-PK3-1 display little change in the four cycles, but it gradually increased in the tests over C-Z5 (Table 3 and Figure 10a–c) because of excessive carbon formation on the surface of the micropore catalyst that blocked the active sites. The TGA analysis of the spent samples (Table 3) reveals that the weight loss of H-Z5-PK3 and H-Z5-PK3-1 is negligible during the four consecutive pyrolysis-regeneration cycles. However, the weight loss of C-Z5 is noticeable. Due to the high acid strength, unique single-crystalline structure, and hierarchical porosities, H-Z5-PK3 exhibits a remarkably improved catalytic lifetime and lower rate of coke formation in the LDPE cracking reaction.

## 4. Conclusions

We have developed a novel strategy to construct aluminum-rich hierarchical ZSM-5 zeolites by using a nitrogen- and halogen-free polymer as a template. The polymer PK3 serves as a dual-function template and facilitates the assembly of the zeolites into single-crystalline form. Moreover, the obtained samples exhibit excellent catalytic performance in LDPE cracking. This improved catalytic performance is attributed to the strong acidity of the ZSM-5 zeolites, as well as the presence of mesopores. The successful application of this polymer template to the construction of aluminum-rich hierarchical ZSM-5 zeolites highlights that this strategy has potential applications in the synthesis of other uniform hierarchical materials.

## Figures and Tables

**Figure 1 nanomaterials-12-01633-f001:**
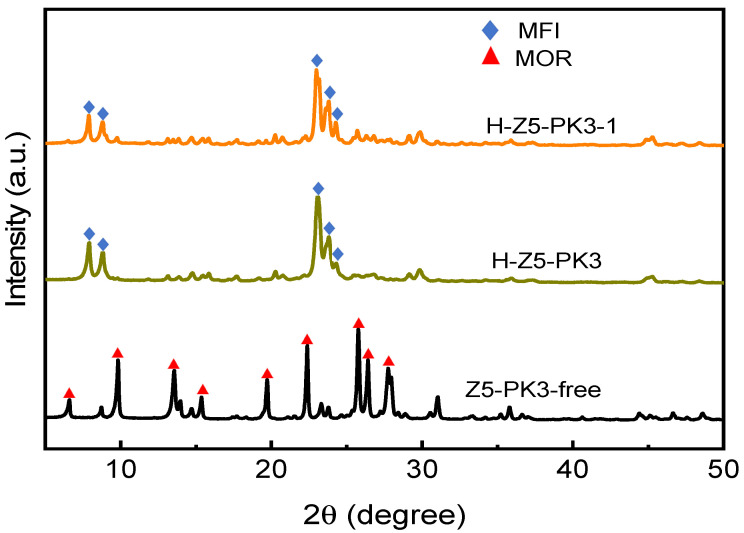
XRD patterns of the different samples.

**Figure 2 nanomaterials-12-01633-f002:**
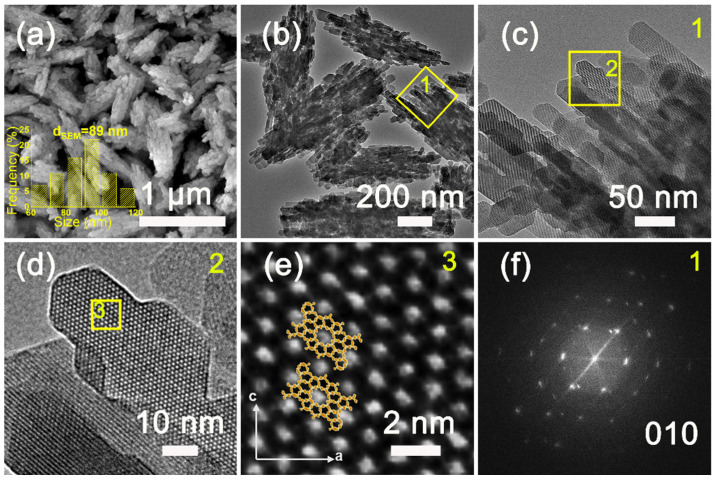
(**a**) SEM and (**b**–**e**) TEM micrographs (inset: size distribution), and (**f**) FFT diffraction pattern of H-Z5-PK3.

**Figure 3 nanomaterials-12-01633-f003:**
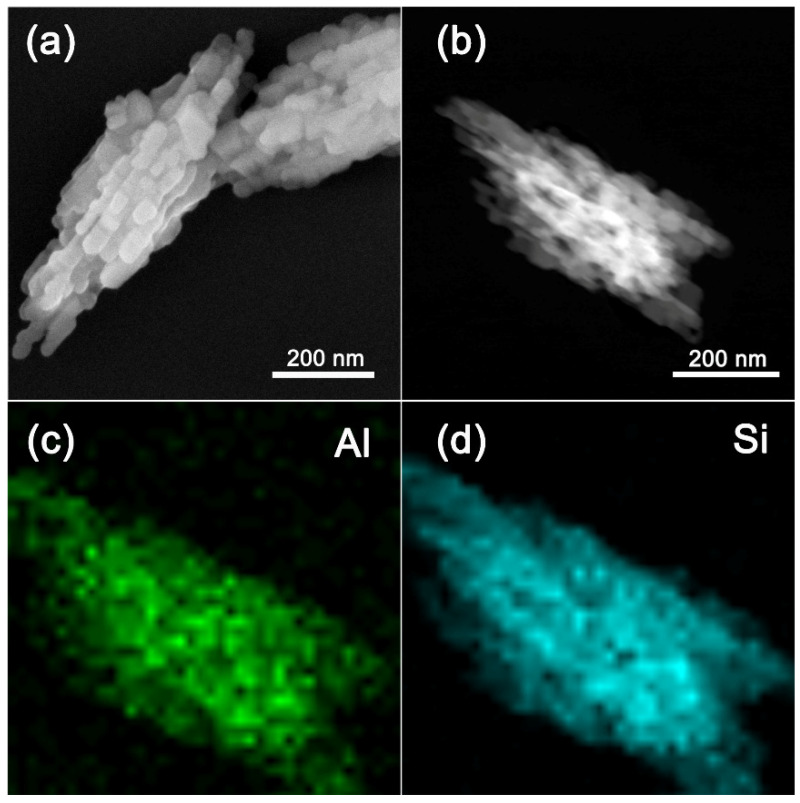
(**a**) SEM and (**b**) HADDF images with respective TEM-EDX elemental mapping (**c**,**d**) of H-Z5-PK3.

**Figure 4 nanomaterials-12-01633-f004:**
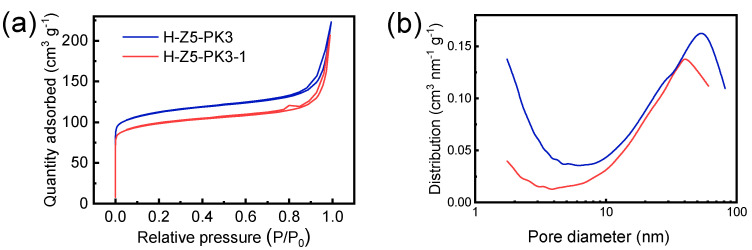
(**a**) Ar adsorption–desorption isotherms and (**b**) BJH pore–size distribution curves of H-Z5-PK3 and H-Z5-PK3-1.

**Figure 5 nanomaterials-12-01633-f005:**
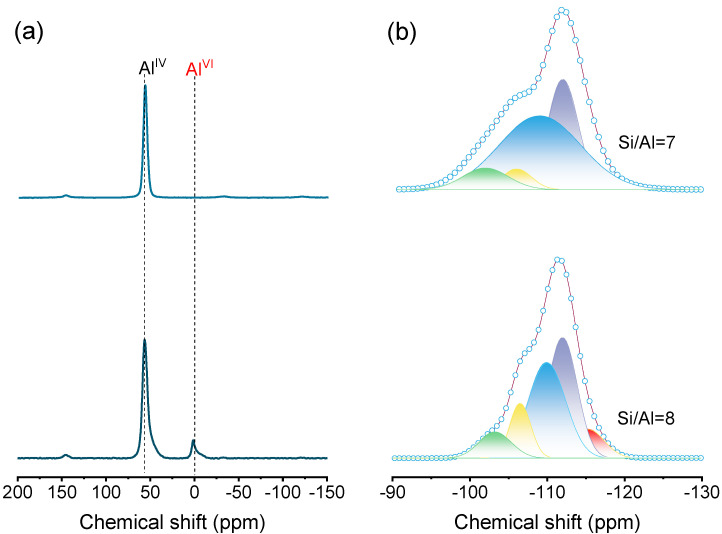
Solid–state (**a**) ^27^Al and (**b**) ^29^Si MAS NMR spectra of Na-Z5-PK3 (**top**) and H-Z5-PK3 (**bottom**).

**Figure 6 nanomaterials-12-01633-f006:**
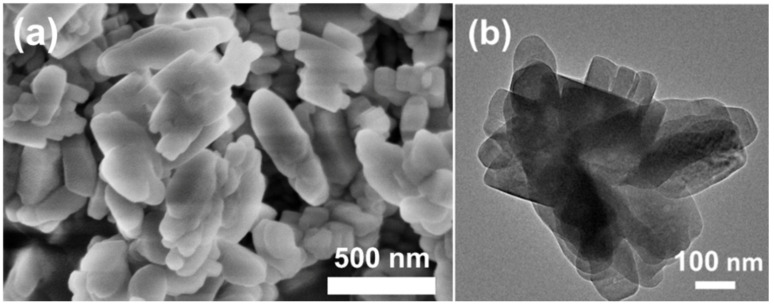
(**a**) SEM and (**b**) TEM micrographs of Z5-PK3-free.

**Figure 7 nanomaterials-12-01633-f007:**
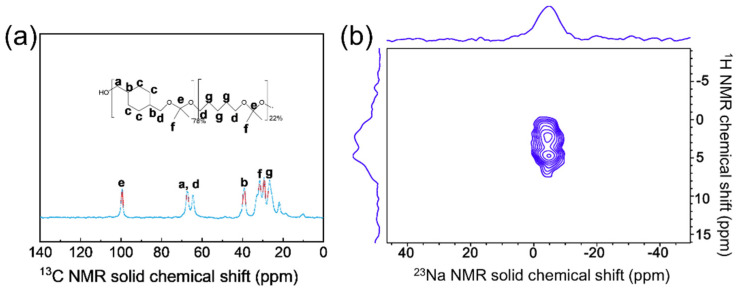
Solid–state (**a**) ^13^C MAS NMR and (**b**) 2D ^23^Na {^1^H} HETCOR NMR spectra of Na-Z5-PK3.

**Figure 8 nanomaterials-12-01633-f008:**
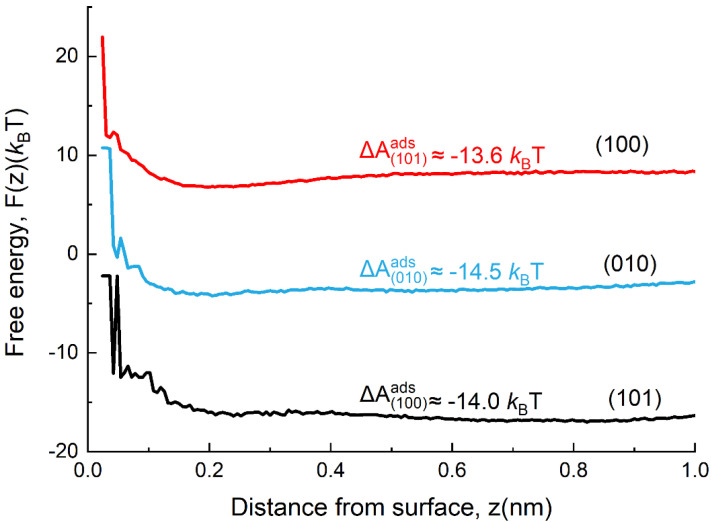
Free energy as a function of the PK3 center–of–mass distance from the (100), (010), and (101) surfaces of ZSM-5 computed using USMD.

**Figure 9 nanomaterials-12-01633-f009:**
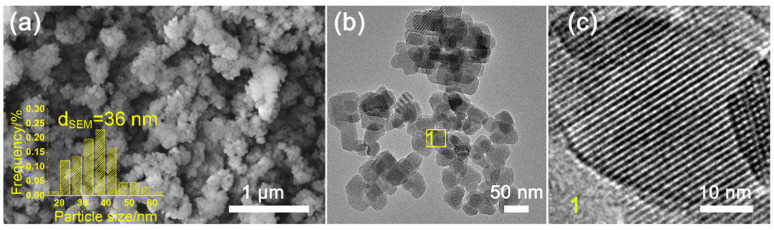
(**a**) SEM and (**b**) low–magnification TEM images of Z5-PK3-1 (inset: the corresponding particle size distribution); and (**c**) high–magnification TEM image of region 1 in (**b**).

**Figure 10 nanomaterials-12-01633-f010:**
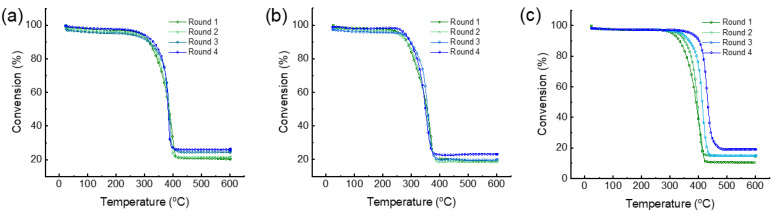
Weight loss curve of LDPE pyrolysis over (**a**) H-Z5-PK3-1, (**b**) H-Z5-PK3, and (**c**) C-Z5 in the four consecutive pyrolysis–regeneration cycles.

**Table 1 nanomaterials-12-01633-t001:** Elemental analyses, textural parameters, and acid amounts of the different ZSM-5 samples.

Sample	Si/Al ^1^	S_BET_ ^2^ (m^2^ g^−1^)	S_micro_ ^3^ (m^2^ g^−1^)	S_meso_ ^3^ (m^2^ g^−1^)	V_total_ (cm^3^ g^−1^)	V_micro_ ^3^ (cm^3^ g^−1^)	V_meso_ ^3^ (cm^3^ g^−1^)	Acid Amounts ^4^ (µmol·g^−1^)
H-Z5-PK3-1	9	397	299	98	0.30	0.16	0.14	355
H-Z5-PK3	8	421	312	109	0.32	0.16	0.16	382
C-Z5	12	345	255	90	0.18	0.15	0.03	293

Notes: ^1^ Measured by ICP; ^2^ total specific surface area calculated by applying the Brunauer–Emmett–Teller (BET) equation using the linear part (0.05 < P/P_0_ < 0.30) of the adsorption isotherm; ^3^ micropore and mesopore specific surface areas, and micropore and mesopore volumes calculated using the *t*-plot method; ^4^ determined by IR spectra of adsorbed pyridine.

**Table 2 nanomaterials-12-01633-t002:** Relative peak areas of the ^29^Si MAS NMR spectra of the Z5-PK3.

δ_Si_ (ppm) ^1^	Q^n^ Environments	Si Distribution (%) ^2^	
		Na-Z5-PK3	H-Z5-PK3
−102	Q^3^ (0Al)	8	9
−106	Q^4^ (1Al)	5	12
−109	Q^4^ (1Al)	51	36
−112	Q^4^ (0Al)	29	35
−115	Q^4^ (0Al)	7	8

Notes: ^1^ Chemical shift after deconvolution; ^2^ calculated from the peak areas of the different δ_Si_ from the ^29^Si MAS NMR profiles in Figure 5.

**Table 3 nanomaterials-12-01633-t003:** Catalytic performance of the different ZSM-5 zeolites for LDPE cracking.

Item	Sample	Round 1	Round 2	Round 3	Round 4
T20 (°C) ^1^	H-Z5-PK3-1	343	349	356	361
	H-Z5-PK3	307	311	319	327
	C-Z5	358	375	396	418
T50 (°C) ^2^	H-Z5-PK3-1	385	383	382	385
	H-Z5-PK3	355	352	350	356
	C-Z5	392	396	413	432
Weight loss (%) ^3^	H-Z5-PK3-1	0.77	0.5	0.48	0.77
	H-Z5-PK3	0.62	0.49	0.48	0.62
	C-Z5	6.1	7.2	6.1	8.3

Notes: ^1^ Temperatures for 20 wt% of LDPE pyrolyzed to hydrocarbons; ^2^ temperatures for 50 wt% of LDPE pyrolyzed to hydrocarbons; ^3^ weight loss of the spent samples after heating from 25 to 700 °C at a ramp rate of 10 °C min^−1^ in O_2_ (90 mL min^−1^).

## Data Availability

Not applicable.

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
