# Peer review of "Nitrogen- and Halogen-Free Multifunctional Polymer-Directed Fabrication of Aluminum-Rich Hierarchical MFI Zeolites"

_nanomaterials, 2022, doi:10.3390/nano12101633_

Round 1

Reviewer 1 Report

Reviewer comments

In this manuscript authors claimed to use an eco-friendly polymer which does not contain nitrogen and halogen elements to synthesize aluminum-rich hierarchical ZSM-5 zeolite with a Si/Al ratio of 8. Authors demonstrated that the nitrogen- and halogen-free polymer play a crucial role in the formation of the ZSM-5 zeolite by serving as a template for constructing the hierarchical micro/mesoporous structure. In addition, authors used synthesized zeolites for catalytic conversion of waste plastic into hydrocarbons.

The reported results are interesting; however, the manuscript needs a major revision to publish in Nanomaterials journal.

Authors could consider the following suggestions, while they perform the revision.

  1. Authors described that the strategies normally employ nitrogen- and/or halogen-containing compounds as structure-directing agents or macro/mesoporogens generating the emissions of toxic NOx and/or HX into the environment during their removal process via high temperature calcination. Therefore, direct synthesis of aluminum-rich hierarchical zeolites via a green route is highly desirable. However, authors used a polymer [(OH-[CH2-C6H10-CH2-O-C(CH3)2-O]78%-[C5H10-O-C(CH3)2-O]22%] with large amount of carbon and it obviously produces CO2, a famous greenhouse gas. Therefore, the justification the authors mentioned is valid.

  1. Authors also mentioned that their work enables the green synthesis of high-quality hierarchical zeolites without requiring quaternary ammonium templates. I disagree with authors statement as the synthesis of polymer require special organic sacrificial reagents such as 1,4-Cyclohexanedimethanol, 1,5-pentanediol, 2,2-dimethoxypropane etc. I believe that this process is not green.

  1. Authors have not evaluated the products formed during the waste plastic, it is important to study this aspect as well.

  1. Authors should identify all the XRD reflections in Fig. 1.

Author Response

General Comment: In this manuscript authors claimed to use an eco-friendly polymer which does not contain nitrogen and halogen elements to synthesize aluminum-rich hierarchical ZSM-5 zeolite with a Si/Al ratio of 8. Authors demonstrated that the nitrogen- and halogen-free polymer play a crucial role in the formation of the ZSM-5 zeolite by serving as a template for constructing the hierarchical micro/mesoporous structure. In addition, authors used synthesized zeolites for catalytic conversion of waste plastic into hydrocarbons. The reported results are interesting; however, the manuscript needs a major revision to publish in Nanomaterials journal. Authors could consider the following suggestions, while they perform the revision.

Authors' Response: Many thanks for Reviewer #1’s positive evaluation on our research and the support for its publication in Nanomaterials. We have provided a detailed point-by-point response to each question.

Comment 1: Authors described that the strategies normally employ nitrogen- and/or halogen-containing compounds as structure-directing agents or macro/mesoporogens generating the emissions of toxic NOx and/or HX into the environment during their removal process via high temperature calcination. Therefore, direct synthesis of aluminum-rich hierarchical zeolites via a green route is highly desirable. However, authors used a polymer [(OH-[CH2-C6H10-CH2-O-C(CH3)2-O]78%-[C5H10-O-C(CH3)2-O]22%] with large amount of carbon and it obviously produces CO2, a famous greenhouse gas. Therefore, the justification the authors mentioned is valid.

Authors' Response: We agree with reviewer #1 that the polymer [(OH-[CH2-C6H10-CH2-O-C(CH3)2-O]78%-[C5H10-O-C(CH3)2-O]22%] (PK3) contains large amount of carbon and CO2 will be produced during its removal process via high-temperature calcination. In fact, high temperature calcination is usually required to remove the template in almost all template strategies, so carbon dioxide emission is inevitable. Additionally, the pollutants NOx or/and HX that are more toxic than CO2 to the human health and environment are also generated by using the conventional templates. Therefore, the emission standard of NOx and HX is much stricter than that of CO2. Fortunately, we developed a nitrogen-/halogen-free template for the synthesis of Al-rich hierarchical MFI zeolite in this study, which can completely avoid the emissions of NOx and HX. To the best of our knowledge, there is no report on the fabrication of Al-rich hierarchical MFI zeolite by using a nitrogen-/halogen-free template. Besides, PK3 is a pH-responsive polymer, i.e., it is stable in basic solution for zeolite synthesis but easily decomposes into its building components including ketone and diols which can be recycled to synthesis of PK3. Based on this, PK3 in the as-synthesized zeolite has a potential to be removed during the transformation of Na-form zeolite into H-form one via a one-step acid exchange without using high-temperature calcination. Because the purpose of this work is to develop a nitrogen-/halogen-free template for the synthesis of Al-rich hierarchical MFI zeolite, PK3 was removed by employing the conventional calcination, but its removal through acid exchange will be carefully studied in our future study. In order to avoid misunderstanding, we modified the statement “green” in the revised manuscript. (The corresponding revisions are colored in red in the top of Page 1, middle of Page 2 and top of Page 15 in the revised manuscript.)

Comment 2. Authors also mentioned that their work enables the green synthesis of high-quality hierarchical zeolites without requiring quaternary ammonium templates. I disagree with authors statement as the synthesis of polymer require special organic sacrificial reagents such as 1,4-cyclohexanedimethanol, 1,5-pentanediol, 2,2-dimethoxypropane etc. I believe that this process is not green.

Authors' Response: As stated in comment 1, PK3 developed in this study is a pH-responsive polymer, i.e., it is stable in basic solution for zeolite synthesis but easily decomposes into its building components including ketone and diols which can be recycled to synthesis of PK3. Based on this, PK3 in the as-synthesized zeolite has a potential to be removed during the transformation of Na-form zeolite into H-form one via a one-step acid exchange without using high-temperature calcination. Because the purpose of this work is to develop a nitrogen-/halogen-free template for the synthesis of Al-rich hierarchical MFI zeolite, PK3 was removed by employing the conventional calcination, but its removal through acid exchange will be carefully studied in our future study. In order to avoid misunderstanding, we modified the corresponding statement in the revised manuscript. (The corresponding revision is colored in red in middle of Page 1 in the revised manuscript.)

Comment 3. Authors have not evaluated the products formed during the waste plastic, it is important to study this aspect as well.

Authors' Response: Thanks for the reviewer's valuable suggestion. LDPE cracking reaction is a typical bulky molecules involved reaction catalyzed by acid catalysts. Recently, Polshettiwar et al. (Nat. Commun. 2020, 11, 3828) demonstrated the acidity and accessible acid sites of the acidic aluminosilicates through the pyrolysis temperature in LDPE pyrolysis and pyrolysis-regeneration cycles. Dusselier et al. (Angew. Chem. Int. Ed. 2021, 60, 24189) also proved that the protonic high-silica FAU zeolites with smaller crystal sizes and unique acid site distributions showed excellent performances in LDPE pyrolysis and consecutive pyrolysis-regeneration cycles in term of pyrolysis temperature. Based on these studies, we carried out LDPE pyrolysis and pyrolysis-regeneration cycles reaction to test the advantage in pore structures and acid strengths of the obtained aluminum-rich hierarchical ZSM-5 zeolite. Our results reflected by pyrolysis temperature have revealed that the obtained zeolite exhibited superior catalytic activity and a remarkably improved catalytic lifetime due to the high acid strength, unique single-crystalline structure, and hierarchical porosities. Therefore, the products formed during the LDPE pyrolysis and pyrolysis-regeneration cycles are not evaluated in our present work, and will be concerned in our further study.

Comment 4. Authors should identify all the XRD reflections in Fig. 1.

Authors' Response: According to the Reviewer #1’s suggestion, we identified all the XRD reflections in Figure 1 in the revised manuscript.

Figure 1. XRD patterns of the different samples.

We thank the reviewer again for the insightful and helpful comments.

Reviewer 2 Report

The article presents the synthesis of the zeolite MFI with unusually high Al content, Si/Al =8. The second notable feature is the structure directing agent, which in this case is a polymer that does not contain heteroatoms except O. The introduction is basically correct except that it is missing some important caveats, which should also be included for the readers benefit, be mentioned and discussed:

1.Yes, it is true that the conventional most active MFI does have Si/Al around 12 and it is not trivial to lower the Si/Al and still get MFI. However, such successful attempts have been made and can be found in the literature. I encourage the authors to comment on that, find and give examples. For instance, US patent 6180550 by J.S. Beck et al, in 2001, reports such a preparation. I am sure there are others.

2.Increasing Al content is one of the methods of enhancing zeolite activity. The second is decreasing the crystal size. The reference ZSM-5 appears to have larger crystals than the reported one so both effects can contribute to the claimed higher activity. This should be pointed out.

The characterization part requires major overhaul with greater attention to details not just the statement of facts that fit the narrative, but can be questioned. The authors should respond and correct them with greater detail and consideration of alternative explanation (NMR, adsorption, IR).

3.Adsorption Figure 4. I disagree with the statement about the observed hystereses: ‘The coexistence of micropores and mesopores of Z5-PK3 can be further confirmed by the Ar adsorption–desorption isotherms, which show the combined features of type I and IV isotherms with a clear H3-type hysteresis loop at P/P0 that ranges from 0.80 to 0.99’. The hysteresis is barely visible and small in comparison to truly micro-mesoporous MFI like in references 15 and 16. In this pressure region hysteresis is mainly due to intercrystalline voids and is not very significant. The pore diameter plot has too much scatter, probably as a result of experimental point fluctuations, and is not very meaningful. Some smoothing is clearly required. Explain comparable external surface areas for the PK and the reference catalyst with much lower mesopore volume of the latter.

4.Al NMR – the authors must confirm and indicate clearly in the article that this corresponds to the calcined sample in the hydrogen form. The lack of Al signal at 0 due to hexagonal Al is astounding, almost incredible for such high Al content and must be explicitly confirmed, maybe with a comment.

5.Si NMR – the profile has very low resolution without any discernible maxima, except the main, so it is possible to fit anything one wants with sufficient number of peaks that can be chosen arbitrarily. The analysis is incomplete and maybe incorrect as it omits the Q3 Si with silanols (not attached to Al) between -95 and -105, and possibly other Si-OH groups. It is very hard to think about ZSM-5 without such Si defects. The presented analysis is too superficial and should include a table with deconvoluted peak positions, intensities, etc.

6.Acid amounts by IR (table 2) – the values given are roughly 10 times smaller than expected for such high Al content if all Al atoms generate acid sites. This data is flawed and needs correction.

7.Catalytic results – explain all symbols in the caption or footnote to the table.

As for mechanism, since no one knows in detail how zeolites are formed, the analysis of possible mechanism is too speculative to spend too much space to discuss. I think this part should be shortened and more space dedicated to confirming the nature of the material (more discussion of characterization) and its merits.

Author Response

General Comment: The article presents the synthesis of the zeolite MFI with unusually high Al content, Si/Al=8. The second notable feature is the structure directing agent, which in this case is a polymer that does not contain heteroatoms except O. The introduction is basically correct except that it is missing some important caveats, which should also be included for the readers benefit, be mentioned and discussed:

Comment 1. Yes, it is true that the conventional most active MFI does have Si/Al around 12 and it is not trivial to lower the Si/Al and still get MFI. However, such successful attempts have been made and can be found in the literature. I encourage the authors to comment on that, find and give examples. For instance, US patent 6180550 by J.S. Beck et al, in 2001, reports such a preparation. I am sure there are others.

Authors' Response: Thank you for Reviewer #2’s valuable suggestion. We have added related literatures in the proper place of the revised manuscript. (The corresponding revision is colored in red in the top of Page 2 in the revised manuscript.)

Comment 2. Increasing Al content is one of the methods of enhancing zeolite activity. The second is decreasing the crystal size. The reference ZSM-5 appears to have larger crystals than the reported one so both effects can contribute to the claimed higher activity. This should be pointed out.

Authors' Response: We strongly agree with Reviewer #2 that decreasing the crystal size can enhance the activity of zeolite, and we revised the corresponding statement in the revised manuscript. (The corresponding revision is colored in red in the top of Page 12 in the revised manuscript.)

Comment 3. Adsorption Figure 4. I disagree with the statement about the observed hystereses: ‘The coexistence of micropores and mesopores of Z5-PK3 can be further confirmed by the Ar adsorption–desorption isotherms, which show the combined features of type I and IV isotherms with a clear H3-type hysteresis loop at P/P0 that ranges from 0.80 to 0.99. The hysteresis is barely visible and small in comparison to truly micro-mesoporous MFI like in references 15 and 16. In this pressure region hysteresis is mainly due to intercrystalline voids and is not very significant. The pore diameter plot has too much scatter, probably as a result of experimental point fluctuations, and is not very meaningful. Some smoothing is clearly required. Explain comparable external surface areas for the PK and the reference catalyst with much lower mesopore volume of the latter.

Authors' Response: It is true that in this pressure region hysteresis is mainly due to the presence of intercrystalline voids and is not very significant as mentioned in the references (Matter 2020, 3(4), 1226-1245; Angew. Chem. Int. Ed. 2020, 59, 19582–19591). According to your valuable suggestion, we have re-written this part. Besides, on account of the pore diameter plot has too much scatter, we have replaced it with BJH pore size distribution curves. (The corresponding revision is colored in the bottom of Page 4 in the revised manuscript, and Figure 4 was updated.)

Figure 4. (a) Ar adsorption–desorption isotherms and (b) BJH pore size distribution curves of H-Z5-PK3 and H-Z5-PK3-1.

Comment 4. Al NMR – the authors must confirm and indicate clearly in the article that this corresponds to the calcined sample in the hydrogen form. The lack of Al signal at 0 due to hexagonal Al is astounding, almost incredible for such high Al content and must be explicitly confirmed, maybe with a comment.

Authors' Response: In our original manuscript, the sample for Al MAS NMR characterization is in Na-form, and only consists of framework Al. However, after ion-exchange and calcination, the Al MAS NMR spectrum of H-form sample (H-Z5-PK3) shows a weak band at 0 ppm relates to extra-framework Al-species (Figure 5a). We have clearly marked the samples and revised the statement in the revised manuscript. (The corresponding revision is colored in red in the middle of Page 7 in the revised manuscript, and Figure 5 was updated.)

Figure 5. Solid-state (a) 27Al and (b) 29Si MAS NMR spectra of Na-Z5-PK3 (top) and H-Z5-PK3 (bottom).

Comment 5. Si NMR – the profile has very low resolution without any discernible maxima, except the main, so it is possible to fit anything one wants with sufficient number of peaks that can be chosen arbitrarily. The analysis is incomplete and maybe incorrect as it omits the Q3 Si with silanols (not attached to Al) between -95 and -105, and possibly other Si-OH groups. It is very hard to think about ZSM-5 without such Si defects. The presented analysis is too superficial and should include a table with deconvoluted peak positions, intensities, etc.

Authors' Response: As for the reviewer #2’s concern, we have indicated a significant difference with different colors in Figure 5b, and added a Table of relative peak areas of the 29Si MAS NMR spectra of the Na- and H-type Z5-PK3 in the revised manuscript. Moreover, the Q3 Si defects is actually existing, and we have modified the sentence according to the comment. (The corresponding revision is colored in red in the middle of Page 7 in the revised manuscript, and Table 2 was added.)

Table 2. Relative peak areas of the 29Si MAS NMR spectra of the Z5-PK3.

δSi (ppm)1

Qn environments

Si distribution (%)2

Na-Z5-PK3

H-Z5-PK3

-102

Q3(0Al)

8

9

-106

Q4(1Al)

5

12

-109

Q4(1Al)

51

36

-112

Q4(0Al)

29

35

-115

Q4(0Al)

7

8

Notes: 1Chemical shift after deconvolution; 2calculated from the peak areas of the different δSi from the 29Si MAS NMR profiles in Figure 5.

Comment 6. Acid amounts by IR (table 2) – the values given are roughly 10 times smaller than expected for such high Al content if all Al atoms generate acid sites. This data is flawed and needs correction.

Authors' Response: As required, we retest the samples and the results are listed in Table 1. The acid amounts of the samples have still not reached the level stated by reviewer #2. This may be caused by the use of different instrument. Even so, the samples in this study were characterized under the same conditions and on the same instrument, we believe that it is reasonable to compare them. (The new data are listed in Table 1.)

Table 1. Elemental analyses, textural parameters and acid amounts of the different ZSM-5 samples.

Sample

Si/Al1

SBET2

(m2 g-1)

Smicro3

(m2 g-1)

Smeso3

(m2 g-1)

Vtotal

(cm3 g-1)

Vmicro3

(cm3 g-1)

Vmeso3

(cm3 g-1)

Acid amounts4 (µmol. g-1)

H-Z5-PK3-1

9

397

299

98

0.30

0.16

0.14

355

H-Z5-PK3

8

421

312

109

0.32

0.16

0.16

382

C-Z5

12

345

255

90

0.18

0.15

0.03

293

Notes: 1Mesuraed by ICP; 2total specific surface area calculated by applying the Brunauer–Emmett–Teller (BET) equation using the linear part (0.05 < P/P0 < 0.30) of the adsorption isotherm; 3micropore and mesopore specific surface areas, and micropore and mesopore volumes calculated using the t-plot method; 4determined by IR spectra of adsorbed pyridine.

Comment 7. Catalytic results – explain all symbols in the caption or footnote to the table.

Authors' Response: As suggested, we have added the explanation of all symbols in the footnote to Table 3. (Table 3 was updated.)

Table 3. Catalytic performance of the different ZSM-5 zeolites for LDPE cracking.

Item

Sample

Round 1

Round 2

Round 3

Round 4

T20 (oC)1

H-Z5-PK3-1

343

349

356

361

H-Z5-PK3

307

311

319

327

C-Z5

358

375

396

418

T50 (oC)2

H-Z5-PK3-1

385

383

382

385

H-Z5-PK3

355

352

350

356

C-Z5

392

396

413

432

Weight loss (%)3

H-Z5-PK3-1

0.77

0.5

0.48

0.77

H-Z5-PK3

0.62

0.49

0.48

0.62

C-Z5

6.1

7.2

6.1

8.3

Notes: 1Temperatures for 20 wt% of LDPE pyrolyzed to hydrocarbons; 2temperatures for 50 wt% of LDPE pyrolyzed to hydrocarbons; 3weight loss of the spent samples after heated from 25 °C to 700 °C at a ramp rate of 10 °C min-1 in O2 (90 mL min-1).

Comment 8. As for mechanism, since no one knows in detail how zeolites are formed, the analysis of possible mechanism is too speculative to spend too much space to discuss. I think this part should be shortened and more space dedicated to confirming the nature of the material (more discussion of characterization) and its merits.

Authors' Response: According to your suggestion, we have shortened this section. (The corresponding revisions are marked from the bottom of Page 8 to the middle of Page 9 in the revised manuscript.)

Round 2

Reviewer 2 Report

The authors’ response is very thorough and detailed. The additions and changes are satisfactory so it can be accepted but there are still minor issues that should be addressed (no need to re-review). The authors incorrectly site the Si/Al ratio in ref. 6 as 15/1. The last value concerns silica/alumina ratio (Si/Al2) so Si/Al is 7.5 and comparable to the authors’ Si/Al = 8. It would not be relevant otherwise and the comment about impurities should be specific (what they are). As a matter of fact, it looks that the sample Z5-PK3-1 has also MOR impurity seen as the small peak near 10 deg. 2-theta. Last comment concerns the particle size distribution – Figure 4b. It has an odd profile but I am not sure how to improve it.

Author Response

General Comment: The authors’ response is very thorough and detailed. The additions and changes are satisfactory so it can be accepted but there are still minor issues that should be addressed (no need to re-review).

Authors' Response: Many thanks for Reviewer’s positive evaluation on our research and the support for its publication in Nanomaterials. We have provided a detailed point-by-point response to each question.

Comment 1: The authors incorrectly site the Si/Al ratio in ref. 6 as 15/1. The last value concerns silica/alumina ratio (Si/Al2) so Si/Al is 7.5 and comparable to the authors’ Si/Al = 8.

Authors' Response: We are sorry for incorrect site the Si/Al ratio in ref. 6. According to your valuable suggestion, we have modified the sentence. (The corresponding revision is colored in red in the top of Page 2 in the revised manuscript.)

Comment 2: It would not be relevant otherwise and the comment about impurities should be specific (what they are). As a matter of fact, it looks that the sample Z5-PK3-1 has also MOR impurity seen as the small peak near 10 deg. 2-theta.

Authors' Response: It is true that there is small part of MOR impurity in Z5-PK3-1. According to your valuable suggestion, we have re-written this part. (The corresponding revision is colored in red in the middle of Page 4 in the revised manuscript.)

Comment 3: Last comment concerns the particle size distribution – Figure 4b. It has an odd profile but I am not sure how to improve it.

Authors' Response: On account of the pore diameter plot has an odd profile, we also found the phenomenon in literature (Figure R1. Fuel 2017, 202, 563–571.). In Figure 4b, although there is a little fluctuant around 29 nm, which maybe result from the existing of minor amount of mesopores at this range, but the mesopore size distribution centered at 50 nm is obvious. 
